# Utility of Cognitive Neural Features for Predicting Mental Health Behaviors

**DOI:** 10.3390/s22093116

**Published:** 2022-04-19

**Authors:** Ryosuke Kato, Pragathi Priyadharsini Balasubramani, Dhakshin Ramanathan, Jyoti Mishra

**Affiliations:** 1Neural Engineering and Translation Labs, Department of Psychiatry, University of California, San Diego, CA 92037, USA; rkato@eng.ucsd.edu (R.K.); dramanathan@health.ucsd.edu (D.R.); jymishra@health.ucsd.edu (J.M.); 2Department of Mental Health, VA San Diego Medical Center, San Diego, CA 92037, USA

**Keywords:** mental health, anxiety, depression, inattention, hyperactivity, EEG, source localization, machine learning, logistic regression

## Abstract

Cognitive dysfunction underlies common mental health behavioral symptoms including depression, anxiety, inattention, and hyperactivity. In this study of 97 healthy adults, we aimed to classify healthy vs. mild-to-moderate self-reported symptoms of each disorder using cognitive neural markers measured with an electroencephalography (EEG). We analyzed source-reconstructed EEG data for event-related spectral perturbations in the theta, alpha, and beta frequency bands in five tasks, a selective attention and response inhibition task, a visuospatial working memory task, a Flanker interference processing task, and an emotion interference task. From the cortical source activation features, we derived augmented features involving co-activations between any two sources. Logistic regression on the augmented feature set, but not the original feature set, predicted the presence of psychiatric symptoms, particularly for anxiety and inattention with >80% sensitivity and specificity. We also computed current flow closeness and betweenness centralities to identify the “hub” source signal predictors. We found that the Flanker interference processing task was the most useful for assessing the connectivity hubs in general, followed by the inhibitory control go-nogo paradigm. Overall, these interpretable machine learning analyses suggest that EEG biomarkers collected on a rapid suite of cognitive assessments may have utility in classifying diverse self-reported mental health symptoms.

## 1. Introduction

Mental health disorders are a leading cause of national and global disability, affecting 1-in-5 Americans and 1-in-4 humans worldwide [1,2]. In the United States alone, the socio-economic burden of mental health is worth hundreds of billions of dollars every year [3]. A general challenge with optimizing treatment for neuropsychiatric disorders is the lack of simple, scalable measurement tools (such as a blood pressure cuff) to identify objective biological markers for treatment [4]. Such tools would allow for potentially a more rapid identification of biological targets for treatment and a way to “measure” whether a particular treatment approach is working. In addition, such tools may allow for a greater patient understanding of how lifestyle and behavioral approaches affect brain activity.

Prior research shows that alterations in cognitive brain functions are associated with common mental disorders [5,6]. Here, we investigated whether an electroencephalography (EEG), captured during a battery of cognitive tasks, is useful in identifying healthy people who are self-reporting symptoms of common mental health disorders. Such predictions can complement symptom classification and diagnosis efforts made using far more expensive tools, such as resting-state neuroimaging data [7,8]. Many studies suggest that the EEG biomarkers for mental health disorders, such as anxiety [9,10,11], depression [12], and inattentive ADHD [13,14]. Some deep neural networks applied to electroencephalography data claim greater than 90% accuracy in terms of predicting mental health symptoms [15,16,17]. Also, traditionally, several studies relate EEG neural markers to cognitive functions, such as attention [18,19,20], inhibitory control [21,22,23], emotion processing [18,24], working memory, and cognitive load [25,26,27,28]. Yet, specific mental health symptoms have not been predicted from the standpoint of multiple neural markers underlying essential cognitive processes.

For this study, we leverage a scalable cognitive brain-mapping platform that we recently developed called the Brain Engagement Platform (BrainE [29]). A total of 97 healthy adults participated in the study, performing the five fundamental cognitive tasks of selective attention, response inhibition, working memory, interference processing, and emotion interference processing, with simultaneous EEG recorded on all tasks. All participants also provided self-reports of anxiety [30], depression [31], inattention, and hyperactivity [32], but none had a formal clinical diagnosis nor were on any medication. Here, we focused our analyses on how to optimally use the EEG data gathered in the context of core cognitive tasks to classify the subjects based on their noted mental health symptoms. We further used these models to reveal which cognitive task constructs and which cortical activation connectivity “hubs” best predict specific mental health behaviors. To the best of our knowledge, this is the first machine learning study designed to analyze the data from 97 healthy adults performing five fundamental cognitive tasks of selective attention, response inhibition, working memory, interference processing, and emotion interference processing, with simultaneous EEG recorded on all tasks. This design enables the investigation of the relationship between the brain patterns underlying the different cognitive processes within the same subject, and the mental health of the subjects.

## 2. Materials and Methods

All data collected for this study have been previously reported [29], and we used this existing dataset for our symptom prediction analyses.

### 2.1. Overview of the Dataset

The study, Balasubramani et al., (2021), used EEG data of 97 healthy adult subjects [29] that completed five cognitive tasks using the BrainE platform. The details of subject characteristics and methods used for data collection and cognitive task-related EEG processing are presented below and are also previously reported [29]. For building the model features, for each subject and each task, we used the task-averaged EEG data around the global peak amplitudes of the evoked signals as identified in three frequency bands—theta (*θ*, 100–300 msecs), alpha (*α*, 100–300 msecs), and beta (*β*, 400–600 msecs)—for 68 cortical regions of interest (ROIs) that were reconstructed as per the Desikan–Killiany atlas [33]. Thus, the size of the whole dataset was *X_i_* ∈ R^68 × 3 × 5^, *i* = 1, …, 97, or in other words, each subject *X_i_* had 68 ROIs × 3 frequencies × 5 tasks = 1020 dimensions. In addition, each subject was binary-labeled for each of four common psychiatric symptoms: anxiety, depression, inattention, and hyperactivity; if any of the four symptom scores exceeded a threshold of 5, then we labeled the subject as ”1” i.e., with symptoms, else ”0” i.e., with no symptoms [30,31,32].

### 2.2. Dataset Acquisition

*Participants*. In the study by Balasubramani et al., (2021), 102 adult human subjects (mean age 24.8 ± 6.7 years, range 18–50 years, 57 females) participated in the *BrainE* neuro-cognitive assessment study. Participants were recruited using IRB-approved on-campus flyers at UC San Diego as well as via the online recruitment forum, ResearchMatch.org, which hosts a registry of research volunteer participants; the advertisement on the Research Match registry was customized for participants in the general San Diego area (within 50 miles of our research location). Overall, ~50% of participants were university affiliates (lab members and university students), whereas the rest were from the general population (i.e., Research Match registry). All participants provided written informed consent for the study protocol (180140) approved by the University of California San Diego Institutional Review Board (UCSD IRB). Participant selection criteria included healthy adult status, i.e., without any current diagnosis for a neuropsychiatric disorder or current/recent history of psychotropic medications or hospitalization within the past 8 weeks. All participants reported normal/corrected-to-normal vision and hearing and no participant reported color blindness. The majority of participants were right-handed (95 of 102). All participants had at least a high-school level education of 16 years; we did not collect information on higher educational qualifications. Five participants were excluded from the study as they had a current diagnosis for a psychiatric disorder and/or current/recent history of psychotropic medications, so a total of 97 subjects were used for the analyses presented in this study.

### 2.3. Experimental Design

*Mental Health Ratings*. As per our previous study, Balasubramani et al., (2021), all participants completed subjective mental health self-reports using standard instruments. Inattention and hyperactivity ratings were obtained using the ADHD Rating Scale (New York University and Massachusetts General Hospital. Adult ADHD-RS-IV with Adult Prompts. 2003: 9–10), anxiety was measured using the Generalized Anxiety Disorder 7-item scale GAD-7 [30], and depression was reported on the 9-item Patient Health Questionnaire (PHQ-9 [31]. We also obtained demographic variables by self-report including, age, gender, race, and ethnicity, and any current/past history of clinical diagnoses and medications.

*BrainE Neuro-Cognitive Assessments*. These assessments were developed and deployed by NEATLabs on the Unity game engine. The Lab Streaming Layer protocol was used to timestamp each stimulus/response event in each cognitive task. Study participants engaged with *BrainE* assessments on a Windows-10 laptop sitting at a comfortable viewing distance. Participants underwent the following cognitive assessment modules that were completed within a 35 min session. Figure 1 shows the stimulus sequence in each task.
*Selective Attention & Response Inhibition*. Participants accessed a game named Go Green modeled after the standard test of variables of attention [34]. In this simple two-block task, colored rockets were presented either in the upper/lower central visual field. Participants were instructed to respond to green-colored rocket targets and ignore, i.e., withhold their response, to distracting rockets of five other isoluminant colors (shades of cyan, blue, purple, pink, orange). The task sequence consisted of a central fixation ‘+’ cue for 500 msec followed by a target/non-target stimulus of 100 msec duration, and up to a 1 s duration blank response window. When the participant made a response choice, or at the end of 1 s in case of no response, a happy or sad face emoticon was presented for 200 msec to signal response accuracy, followed by a 500 msec inter-trial interval (ITI). To reinforce positive feedback for fast and accurate responding, within 100–400 msec two happy face emoticons were simultaneously presented during the feedback period. Both task blocks had 90 trials lasting 5 min each, with target/non-target trials shuffled in each block. A brief practice period of 4 trials preceded the main task blocks. Summary total block accuracy was provided to participants at the end of each block as a series of happy face emoticons (up to 10 emoticons) in this and in all assessments described below. In the first task block, green rocket targets were sparse (33% of trials), hence selective attention was engaged as in a typical continuous performance attention task. In the second block, green rocket targets were frequent (67% of trials), hence participants developed a prepotent impulse to respond. As individuals must intermittently suppress a motor response to sparse non-targets (33% of trials), this block provided a metric of response inhibition.*Working Memory*. Participants accessed a game named Lost Star that is based on the standard visuo-spatial Sternberg task [35]. Participants were presented with a set of test objects (stars); they were instructed to maintain the visuo-spatial locations of the test objects in working memory for a 3 s delay period, and then responded whether a probe object (star) was or was not located in the same place as one of the objects in the original test set. We implemented this task at the threshold perceptual span for each individual, i.e., the number of star stimuli that the individual could correctly encode without any working memory delay. For this, a brief perceptual thresholding period preceded the main working memory task, allowing for an equivalent perceptual load to be investigated across participants. During thresholding, the set size of the test stars was progressively increased from 1 to 8 stars based on accurate performance; 4 trials were presented at each set size and 100% performance accuracy led to an increment in set size; <100% performance led to one 4-trial repeat of the same set size and any further inaccurate performance aborted the thresholding phase. The final set size at which 100% accuracy was obtained was designated as the individual’s perceptual threshold. Post-thresholding, the working memory task, consisted of 48 trials presented over 2 blocks. Each trial initiated with a central fixation ‘+’ for 500 msec followed by a 1 s presentation of the test set of star objects located at various positions on the screen, then a 3 s working memory delay period, followed by a single probe star object for 1 s, and finally a response time window of up to 1 s in which participants made a yes/no choice whether the probe star had a matching location to the previously presented test set. A happy/sad face emoticon was used to provide accuracy feedback for 200 msec followed by a 500 msec ITI. Summary accuracy was also shown between blocks. The total task duration was 6 min.*Interference Processing*. Participants accessed a game named Middle Fish, an adaptation of the Flanker task [36], which has been extensively used to study interfering/distractor processing. Participants were instructed to respond to the direction of a centrally located target (middle fish) while ignoring all flanking distractor fish. In congruent trials the flanker fish faced the same direction as the central fish, whereas in incongruent trials they faced the opposite direction. A brief practice of 4 trials preceded the main task of 96 trials presented over two blocks for a total task time of 8 min. 50% of trials had congruent distractors and 50% were incongruent. To retain attention, the array of fish was randomly presented in the upper or lower visual field in an equivalent number of trials. In each trial, a central fixation ‘+’ appeared for 500 msec followed by a 100 msec stimulus array of fish and up to a 1 s response window in which participants responded left/right as per the direction of the middle fish. Subsequently a happy/sad face emoticon was presented for 200 msec for accuracy feedback followed by a 500 msec ITI. Summary accuracy was shown between blocks and the total task duration was 8 min.*Emotional Interference Processing*. We embedded this task in the BrainE assessment suite given ample evidence that emotions impact cognitive control processes [37,38,39]. Participants accessed a game named Face Off, adapted from prior studies of attention bias in emotional contexts [40,41,42]. We used a standardized set of culturally diverse faces from the Nim-Stim database for this assessment [43]. We used an equivalent number of male and female faces, each face with four sets of emotions, either neutral, happy, sad, or angry, presented in equivalent number of trials. An arrow was superimposed on the face in each trial, occurring either in the upper or lower central visual field in an equal number of trials, and participants responded to the direction of the arrow (left/right). Participants completed 144 trials presented over three equipartitioned blocks with a shuffled but equivalent number of emotion trials in each block; a practice set of 4 trials preceded the main task. Each trial was initiated with a central fixation ‘+’ for 500 msec followed by a face stimulus with a superimposed arrow of 300 msec duration. As in other tasks, participants responded within an ensuing 1 s response window, followed by a happy/sad emoticon feedback for accuracy (200 msec) and a 500 msec ITI. Summary block accuracy feedback was provided, and the total task duration was 10 min.

**Figure 1 sensors-22-03116-f001:**
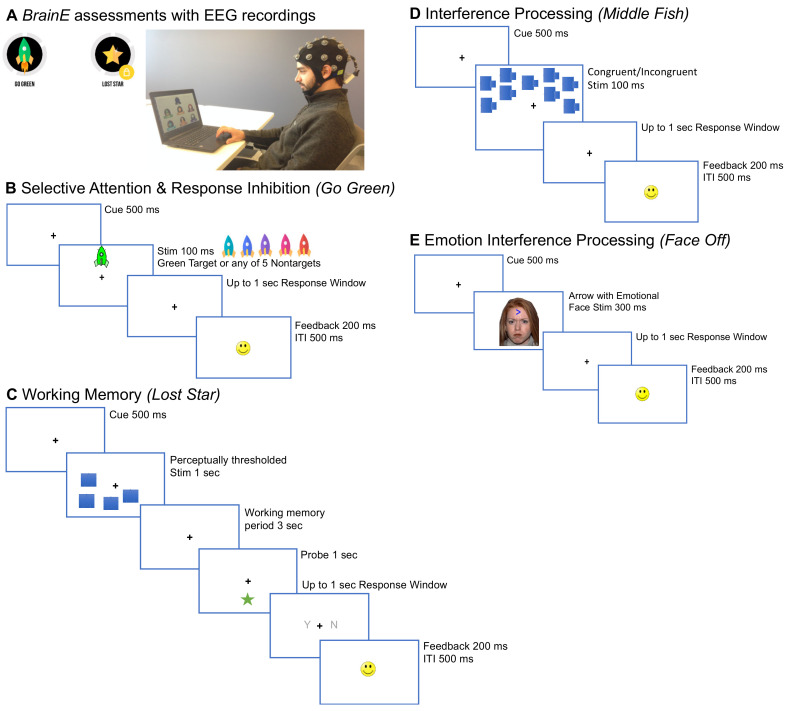
Cognitive assessments delivered on the BrainE platform (adapted from [12]). (**A**) BrainE assessment dashboard with the wireless EEG recording setup. (**B**) The selective attention and response inhibition tasks differ only in the frequency of targets; sparse 33% targets appear in the Selective Attention block and frequent 67% targets appear in the Response Inhibition block. (**C**) Working memory task with perceptually thresholded stimuli. (**D**) Flanker interference processing task; flanking fish may either face the same direction as the middle fish in congruent trials, or the opposite direction in incongruent trials. (**E**) Emotion interference task presents neutral, happy, sad, or angry faces superimposed on an arrow.

### 2.4. Neural Processing Methods

We obtain the processed data published in the Balasubramani et al., (2021) study for training our machine learning model. In the study by Balasubramani et al., (2021), the authors use the Parametric Empirical Bayes (PEB) toolbox [44,45], which makes use of a library of artifacts for cleaning and source localization purposes. A pipeline followed for EEG processing and that study is presented in Appendix A.

Neural Analyses. The study by Balasubramani et al., (2021), applied a uniform processing pipeline to all EEG data acquired simultaneously to the cognitive tasks. This included: (1) data pre-processing, and (2) cortical source localization of the EEG data filtered within relevant theta, alpha, and beta frequency bands.

Data preprocessing was conducted using the EEGLAB toolbox in MATLAB. EEG data were resampled at 250 Hz and filtered in the 1–45 Hz range to exclude ultraslow DC drifts at <1 Hz and high-frequency noise produced by muscle movements and external electrical sources at >45 Hz. We performed 827-point bandpass, zero phase, filtering with transition bandwidth 4.063 Hz, and passband edges of [1, 45] Hz, to enable cleaning of the epoched data of time length [−1.5, 1.5] s; [3, 7] Hz for theta-specific filtering, [8, 12] Hz for alpha-specific, and [13, 30] Hz for beta-specific data analysis. EEG data were average referenced and epoched to relevant stimuli in each task, as informed by the LSL timestamps. Although 24 channels is not a dense set, they are far enough from each other that no common neural signature is removed, but only common-in-phase noise present in all channels is canceled during average referencing [46]. Any task data with missing LSL markers (1.4% of all data) had to be excluded from neural analyses. Any missing channel data (channel F8 in 2 participants) was spherically interpolated to nearest neighbors. Epoched data were cleaned using the autorej function in EEGLAB to remove noisy trials (>5 sd outliers rejected over max 8 iterations; 6.6 ± 3.4% of trials rejected per participant). EEG data were further cleaned by excluding signals estimated to be originating from non-brain sources, such as electrooculographic, electromyographic, or unknown sources, using the Sparse Bayesian learning (SBL) algorithm [45] implemented within the PEB toolbox, (https://github.com/aojeda/PEB, accessed on 12 July 2019) as explained below.

Cortical source localization was performed to map the underlying neural source activations for the ERSPs using the block-Sparse Bayesian learning (SBL) algorithm [44,45] implemented in a recursive fashion in the PEB toolbox. All the ERSP results have been previously published in Balasubramani et al., (2021); these results are shown in Appendix B. SBL is a two-step algorithm in which the first step is equivalent to low-resolution electromagnetic tomography (LORETA [47]). LORETA estimates sources subject to smoothness constraints, i.e., nearby sources tend to be co-activated, which may produce source estimates with a high number of false positives that are not biologically plausible. To guard against this, SBL applies sparsity constraints in the second step wherein blocks of irrelevant sources are pruned. Source space activity signals were estimated and then their root mean squares were partitioned into (1) regions of interest (ROIs) based on the standard 68 brain region Desikan–Killiany atlas [33] using the Colin-27 head model [48] and (2) artifact sources contributing to EEG noise from non-brain sources such as electrooculographic, electromyographic, or unknown sources; activations from non-brain sources were removed to clean the EEG data. The SBL GUI accessible through EEGLAB provides access to an EEG artifact dictionary; this dictionary is composed of artifact scalp projections and was generated based on 6774 ICs available from running Infomax ICA on two independent open-access studies (http://bnci-horizon-2020.eu/database/data-sets, study id: 005-2015 and 013-2015, accessed through PEB toolbox on 12 July 2019). The k-means method is used to cluster the IC scalp projections into Brain, EOG, EMG, and Unknown components. We checked visually that EOG and EMG components had the expected temporal and spectral signatures according to the literature [49]. The SBL algorithm returns cleaned channel space EEG signals in addition to the derived cortical source signals as outputs.

Also, with regards to source localization of the limited number of channels, we note that the 24 channels were placed at standard electrode locations with whole-head coverage as per the 10–20 electrode-placement system. The sparse Bayesian learning (SBL) algorithm that we use for source localization applies sparsity constraints wherein blocks of irrelevant sources are pruned. Notably, this data-driven sparsity constraint reduces the effective number of sources considered at any given time as a solution, thereby reducing the ill-posed nature of the inverse mapping. Thus, it is not that only higher channel density data can yield source solutions [50], but the ill-posed inverse problem can also be solved by imposing more aggressive constraints on the solution to converge on the source model at lower channel densities, as also supported by prior research [51,52]. Furthermore, the SBL algorithm has been benchmarked to produce evidence-optimized inverse source models at 0.95 AUC relative to the ground truth, verified using both data and simulations [44,45]. We have also shown that cortical source mapping with this method has high test-retest reliability [29].

In this study, we first applied SBL to the epoched channel EEG signals; activations from artifact sources contributing to EEG noise, i.e., from non-brain sources such as electrooculographic, electromyographic, or unknown sources, were removed to clean the EEG data. Cleaned subject-wise trial-averaged channel EEG data were then specifically filtered in theta (3–7 Hz), alpha (8–12 Hz), and beta (13–30 Hz) bands and separately source localized in each of the three frequency bands and in each task to estimate their cortical ROI source signals. The source signal envelopes were computed in MATLAB (envelop function) by a spline interpolation over the local maxima separated by at least one time sample; we used this spectral amplitude signal for all neural analyses presented here. We focused on post-stimulus encoding in the 100–300 msec range for theta and alpha bands, and 400–600 msec spectral amplitude range for the beta band signals. These epoch windows were chosen based on the peak global activity of the task-averaged signals in the respective frequency bands. The peak global activations were further baseline-corrected for baseline activity between −750 to −550 msecs, and the resulting epoched signals were used for this study. The source localization results for this dataset have been previously studied in Balasubramani et al., (2021). We present statistically corrected source localization results from Balasubramani et al., (2021) in Appendix C.

### 2.5. Predicting Mental Health Symptoms Using Logistic Regression

In this study, we classified subjects’ mental health self-report scores using the logistic regression command in Jupyter notebook (Python). Specifically, our task was to predict binary [0, 1] labels for all 4 symptoms for each subject. Notably, though we also explored other machine learning algorithms such as random forest, support vector machine, or simple neural network, we found that logistic regression performance was as good as that for the more complex algorithms.

To achieve high accuracy, we modeled the original dataset as well as tested application of data augmentation, feature selection, and oversampling techniques to the original dataset. Data augmentation and feature selection are methods that enhance the number of features (=1020 in the original dataset), whereas oversampling improves the number of samples (=97 in the original set).

### 2.6. Logistic Regression

We used logistic regression with L2 regularization term, which minimizes the following objective function
∑ilog(1+exp(−wTXi.yi))+λwTwminw
where *X* is a matrix whose columns correspond to data points, *y* is a vector of labels, *w* is a vector of coefficients of the logistic regression, *X_i_* represents the *i*-th column of *X*, and *λ* > 0 is a regularization parameter to determine how much to penalize according to the magnitude of *w*. We set *λ* = 1 in all experiments. The L2 regularization term in the model is $\lambda w^{T}w=\lambda ||w|{|}_{2}^{2}$. This model is very similar to *ridge regression* [53], which is a linear regression with L2 regularization term. Given the L2 regularization term the following hold: a.The model can mitigate overfitting or accidental fitting [54]. Furthermore, we apply stratified cross validation during model testing, which also helps to avoid overfitting.b.The model can mitigate the error of estimated coefficients and make logistic regression predictions despite multicollinearity. The variance of coefficients and prediction error of ridge regression is smaller than that of simple regression even if multicollinearity, i.e., a state in which multiple features are strongly correlated, occurs [55]. Therefore, we believe the model is able to stably predict even if many features are correlated.

### 2.7. Data Augmentation

It is common to apply some operation to the original dataset in order to augment its dimension to detect the hidden structure of the dataset [56]. For example, by introducing an intersection term, i.e., the product of the two features *x_i_x_j,_* into the predictive model, we can predict objective value *y* more accurately if the effect of feature *x_i_* on objective value *y* is dependent on another feature *x_j_* [57]. First, we augmented the dimensions of the dataset by computing additional features through (i) statistical measurements, (ii) products of two arbitrary features, and (iii) logarithms of each feature. We added these additional features to the original dataset as new features and, consequently, the number of features increased considerably. A major motivation for conducting data augmentation is to use interactions between features as a separate predictor. By introducing a product of two features (e.g., *x_i_x_j_*), we can compute the importance of the connectivity of two the features in mental health prediction.

#### 2.7.1. Statistical Measures

As shown in Figure 2, we computed seven statistical measures (max(·), min(·), max(·)-min(·), mean(·), std(·), quantile0.75(·), and quantile0.25(·)) across three categories: 68 ROIs, 3 frequency bands (θ, α, β), and 5 cognitive tasks. Namely, we obtained 7 above-mentioned statistical measures × 5 tasks × 3 frequency bands = 105 new features by computing these seven measures across mean of all ROIs. Similarly, we generated 7 measures × 68 ROIs × specific for 3 frequency bands and mean across all tasks = 1428 and 7 measures × 68 ROIs × 5 tasks and generalized across-frequency bands = 2380 features by the measures across frequency band and task, respectively. Consequently, the number of features was enhanced to base features (viz. 68 * 3 frequency bands * 5 tasks = 1020) + 105 + 1428 + 2380 = 4933.

We computed seven statistical measures (max(·), min(·), max(·)-min(·), mean(·), std(·), quantile0.75(·), and quantile0.25(·)) across three categories: 68 ROIs, 3 frequency bands (θ, α, β) and 5 cognitive tasks i.e.,
(1)we obtained 7 × 5 × 3 = 105 new features by computing these seven measures across ROIs.(2)we generated 7 × 68 × 3 = 1428 features by the measures across frequency band.(3)we generated 7 × 68 × 5 = 2380 features by the measures across task, respectively.

#### 2.7.2. Product of Feature Pairs

Then, we computed the product of any two randomly picked pair of features:(1)$$X_{i,j}=X_{i}.X_{j}\;\mathrm{where}\;i\neq j,$$
and added the values as new features. After adding the products of these feature pairs, the number of features became (features till now = 4933) + 4933C2 = 12,169,711.

#### 2.7.3. Log Transform

Finally, we computed the logarithm of the whole dataset obtained so far, which can enhance subtle details in the classification step [58]. Since the domain of logarithms contains only positive real numbers, we calibrated the value of each feature before computing its logarithm. Namely, we computed
(2)$$X_{i,\log}=\log\left(1+\frac{X_{i}-\min_{i}X_{i}}{\max_{i}X_{i}-\min_{i}X_{i}}\right)\;\mathrm{where}\;i=1,2,\ldots 96$$
for each feature and added it to the given features. Since we add the same number of log features to the existing non-log features, the total number of features is now 12,169,711 × 2 = 24,339,422.

### 2.8. Feature Selection

Next, we selected a subset of the 24,339,422 augmented feature set and input these as our predictors for the logistic regression model for predicting symptom scores. We selected this subset of features based on the chi-squared statistic as a criterion to distinguish the “best predictive” features. For this, we computed the chi-squared statistic [59] for each of the augmented features and selected the 40,000 best features among them based on the values of the chi-squared statistic (Figure 3). We chose 40,000 as the number of features selected as we found that sufficiently high accuracy of ~80% is achieved using this feature subset. With regards to highly correlated features, by introducing new xixj features in the data augmentation step, the value of chi-square statistic is the same whether there exists correlation to other features i.e., it is independent of other features, and thus we can use it as a metric to measure feature importance. Note that each accuracy score in Figure 3 is the mean value of 5 iterations of 5-fold stratified cross validation, that is, we conduct 5 × 5 = 25 experiments in total for every number of features.

The chi-squared statistic is a generic measure of the co-occurrence between a feature and an objective nominal variable. It essentially computes how much a neural feature *t* contributes to the classification performance. The chi-squared statistic is computed as follows for each feature:(3)$$\chi^{2}(D,c)=\sum_{e_{c}\in \{0,1\}}\frac{{\left(S_{e_{c}}-E_{e_{c}}\right)}^{2}}{E_{e_{c}}}$$
where *e_c_* = 1 is a variable such that *e_c_* = 1 if the subject belongs to class *c* or *e_c_* = 0 otherwise. *S* is the sum of values of features in class *c* and *E* is the expected sum of the class. For example, if there are three subjects with objective variable *c* = 1, 0, 1 where *c* = 1 if his/her score is above the threshold for a mental health symptom and *c* = 0 otherwise, and their corresponding values of some features are 30, 20, and 40. Then, *S_e_**_c_* and *E_e_**_c_* are computed as follows,
$$\begin{array}{l} S_{e_{0}}=20\\ S_{e_{1}}=30+40=70\\ E_{e_{0}}=(S_{e_{0}}+S_{e_{1}}).P_{e_{0}}=(20+70).\frac{1}{3}=30\\ E_{e_{1}}=(S_{e_{0}}+S_{e_{1}}).P_{e_{1}}=(20+70).\frac{2}{3}=60 \end{array}$$
where *P_e_**_c_* denotes probability that an example belongs to class *c*. Thus, the value of *χ*^2^ becomes
$$\chi^{2}=\frac{{\left(20-30\right)}^{2}}{30}+\frac{{\left(70-60\right)}^{2}}{60}=5$$

If the feature is correlated to the objective variable, the difference of *S_e_**_c_* to *E_e_**_c_* is larger and, consequently, *χ*^2^(D, *c*) becomes larger. Therefore, we can use *χ*^2^(D, *c*) to estimate how much the objective nominal variable depends on the given feature. Note that although the chi-squared statistic was originally defined between categorical dependent and independent variables, it can also be extended to non-negative continuous variables such as peak amplitude of EEG data. The definition shown above is the extended version applied to continuous features.

### 2.9. Oversampling

To mitigate the drawbacks of a small sample size of 97 study participants, we used two methods: (1) SMOTE (Synthetic Minority Oversampling TEchnique), and (2) adding Gaussian noise. Note that we applied these oversampling methods to training data and not to test data, during each of the 5 iterations of 5-fold stratified cross validation.

#### 2.9.1. SMOTE

SMOTE is a method to create new data points by interpolation, and it is widely used for imbalanced data (i.e., a dataset in which the numbers of labels are not equal). SMOTE creates new data points until the numbers of two labels for classification are equal.

#### 2.9.2. Adding Gaussian Noise

We increased the number of samples by adding Gaussian white noise to the original dataset.
(4)$$X_{new}=\overset{-}{X}+\varepsilon ,\;\;\;\;\varepsilon ∼\;N(0,\Sigma )$$

We set Σ as a diagonal matrix such that its diagonal entries are equal to the diagonal entries of _100_^1^ Σ_0_, where Σ_0_ is the covariance matrix of {*X_i_*} and all non-diagonal entries are zero. We generated a nine-times-larger set of new data by adding Gaussian noise to the original set and adding the new data to the original data.

Thus, overall, we opted for the above methodology to construct a simple logistic regression model with the augmented and selected features rather than a complex model with the original features. Although it may be possible to construct some end-to-end deep learning models to achieve good accuracy using only the original feature set, in general, this is very time-intensive, and needs plenty of computational resources to find the appropriate architecture and parameters of neural networks among various models. In addition, it is often difficult to interpret the results achieved by such complex neural nets. Hence, here we opted to use basic statistical measures (max, min, …) and log or polynomial transforms to generate new features because they are usually good starting points to find meaningful features [60,61]. It is true that no new information is added by such feature augmentation. However, by transforming each data point to higher-dimension space by feature augmentation, it becomes easier to find a linear hyperplane that separates all data points appropriately.

### 2.10. Evaluation

#### 2.10.1. Stratified Cross-Validation

For each mental health symptom, we conducted five-fold stratified cross-validation. Namely, we divided the whole dataset into five parts so that the ratio of labels in each part is approximately equal to the ratio of labels in a complete dataset. Then, we assigned test data to one of the five parts while treating the other four as training data and taking averages of the performances of all five patterns.

#### 2.10.2. Sensitivity and Specificity

We used sensitivity and specificity as metrics to evaluate four binary classification problems. Sensitivity and specificity are defined as
(5)$$\mathrm{sensitivity}=\frac{TP}{TP+FN}$$
(6)$$\mathrm{specificity}=\frac{TN}{TN+FP}$$
where *TP* is true-positive, *TN* is true-negative, *FP* is false-positive, and *FN* is false-negative. We determine a subject is true-positive for each symptom if his/her self-report score is no less than 5 or otherwise true-negative as seen in the Dataset Section 2.1.

### 2.11. Assessing ”Hub-like” Spectral Activations That Predict Mental Health Symptom Scores

Many combinations of two features that have different frequency band, task, and statistical measurement attributes belong to the same pair of ROIs. For example, the product of feature 1 (*α*, Face Off task, ROI 19) and feature 2 (*β*, Go Green 2 task, ROI 41) and the logarithm of the product of feature 1 (*β*, Go Green 1 task, ROI 19) and feature 2 (mean across frequency bands, Go Green 1 task, ROI 41) belong to the same ROI pair (19, 41), and, likewise, many other products and logs of product features belong to the same ROI pair (19, 41). To aggregate these values to compute the “strength” of the connection of arbitrary pairs, we used a simple arithmetic sum of chi-squared statistics for each combination,
(7)$$strength_{i_{1},i_{2}}=\sum_{j_{1},j_{2},k_{1},k_{2}}\chi^{2}(i_{1},i_{2},j_{1},j_{2},k_{1},k_{2},l)\;\mathrm{where}\;i_{1}<i_{2}$$
where each notation represents as follows: *i*_1_ and *i*_2_: ROI; *j*_1_ and *j*_2_: frequency band; *k*_1_ and *k*_2_: task; and *l* ∈ {0, 1} is a binary variable that is 1 if the combination is log-transformed or 0 otherwise. Namely, strength*_i_*_1,*i*2_ is the sum of the chi-square statistics of features consisting of the product of all possible pairs of two features and with/without logarithm transform.

We use the same principles for constructing the graphs. Using the “strength”, we can create an edge *e_i_*_1,*i*2_ between two vertices (ROIs) *i*_1_ and *i*_2_ only if strength*_i_*_1,*i*2_ value exceeds some threshold. In our results section, graphs created in the case of a high threshold of 100 are shown.

As the “strength” of the given pair consists of various kinds of combinations, it is also worth analyzing which factor (frequency band, task, etc.) most contributes to the “strength”. Thus, for each pair of vertices (ROIs) and each mental health symptom, the highly contributing feature (belonging to the frequency band and cognitive task) was computed. Namely, the most representative factor of pair (*i*_1_, *i*_2_) was computed as below by finding the max along one feature type and averaging across the other feature types. To compute the max ROI we average the task and frequency band feature types.
(8)$$y^{*}{}_{i_{1},i_{2}}=\underset{x}{\arg\max}\sum_{x}\chi^{2}(i_{1},i_{2},j_{1},j_{2},k_{1},k_{2},l)\;\mathrm{where}\;i_{1}<i_{2}$$
where *x* is a feature of interest, such that *x* ∈ {*θ*, *α*, *β*} if we want to find which frequency band on each edge, or *x* ∈ {1, 2, 3, 4, 5} if we are interested in which task is the dominant feature.

### 2.12. Current Flow Centralities

We further computed current flow closeness and betweenness centralities [62] to measure the extent to which a given ROI, or vertex in a graph, has the largest chi-square value, similar to a “hub” in a network, among all ROIs. Current flow closeness and betweenness centralities are the generalized versions of closeness and betweenness centralities, respectively. We used these current flow versions instead of the standard ones because the latter indicates just the shortest path, whereas the “current flow” centralities also account for paths other than the shortest path.

## 3. Results

### 3.1. Prediction of Mental Health Symptoms

We found that logistic regression performance was best when we used oversampling (OS), data augmentation (DA), and feature selection (FS) of the top contributing features (40,000) from our augmented data, to predict mental health symptoms (Figure 4). We found >80% sensitivity and specificity, particularly for anxiety and inattention, when we employed data augmentation, feature selection, and oversampling techniques to the data. When assuming that the expected proportions by chance are 50% for each class label, the 95th percentile confidence intervals based on the standard error of the proportion are 50 ± 9.95%. For more robust control, we performed 100 iterations of random shuffling of class labels and thereby their association with the samples and performed a stratified cross-validation procedure (see Section 2.10) to find the 95th percentile randomized control accuracy distributions for anxiety, depression, inattention, and hyperactivity; these were 0.60, 0.63, 0.59, and 0.59, respectively. The actual mean ± std accuracies for the DA + FS + OS models were found to be above these random control thresholds, and they were 0.86 ± 0.05, 0.78 ± 0.06, 0.85 ± 0.10, and 0.73 ± 0.09, respectively suggesting the statistical significance of our results. In particular, we show sensitivity and specificity results in Figure 4.

### 3.2. Different “Hub-like” Spectral Activations during Cognitive Tasks Predict Mental Health Symptoms

Next, we investigated the neural features that contribute to the prediction of mental health symptoms. More precisely, we identified “hub-like” spectral features in a graph consisting of 68 vertices, corresponding to the 68 ROIs, by mapping the chi-square statistic of pairs of ROIs to the graph. As shown in Table 1, for anxiety, we found that the representative 40,000 selected features contain largely the products or logarithms of the products of a specific pair of features (as generated through data augmentation), whereas the chi-squared statistics, i.e., the rank of single features, tended to be much lower. In addition, many of these high-rank pairs consisted of different ROIs, such as (19, 41) or (7, 43) (Table 1). Therefore, we present these connections among ROIs as a graph for better visual interpretation. Note that we used the chi-square statistic of not only the best 40,000 features used for classification, but also of all 2,494,820 features to precisely compute connection strengths (as below).

We represent the cumulative strengths of chi-square statistics (Equation (7)) corresponding to the 68 ROIs in the form of node sizes of a topographical plot, where the locations of the ROI nodes in the topographical plot are illustrative of their actual 2D brain locations as presented in the Desikan–Killiany atlas. We then examine which ROIs, frequency bands, and tasks contribute most significantly to symptom prediction with the highest chi-squared values.

As shown in Figure 5, a small number of graph vertices (ROIs) dominate for predicting each mental health symptom. Table 2 shows these high-strength or highest-contributing vertices (ROIs) and their corresponding brain regions.

Further, in Figure 6 and Figure 7, we detail results by frequency band and type of task, respectively. These graphs show the cumulative chi-square strength values (Equation (7)) as color-coded lines for the top 5 ROIs. In the case of the Figure 6 plots that are color-coded by frequency band, we observe that α (blue), θ (green), β (red), both θ and α bands are dominant in anxiety, depression, hyperactivity, and inattention. In Figure 7, in the visualization by cognitive task type, MiddleFish (interference processing, blue) is the highest contributor to predicting anxiety; MiddleFish (interference processing, blue), FaceOff (emotional interference processing, purple), and GoGreen I (go-nogo selective attention, red) are the top contributors for the prediction of depression; MiddleFish (interference processing, blue) and FaceOff (emotional interference processing, purple) are the top contributors for the prediction of inattention; and GoGreen I (go-nogo selective attention, red) and Go Green II (go-nogo response inhibition, yellow) are the top predictors for hyperactivity.

## 4. Discussion

In this study, we performed logistic regression-based prediction analyses of neural spectral activations that were simultaneously recorded while adult participants engaged in a suite of cognitive task assessments, in order to predict mental health symptoms. Though many deep neural networks are gaining momentum for understanding biomarkers of psychiatric disorders [15,16,17], there is a school of thought that favors the development of interpretable machine learning models [63,64,65,66], instead of deep learning black-box models, to gain insights into clinical translation. In our study, with the help of simplistic regression methods and guided feature augmentation, we propose that our findings are more interpretable and traceable to the neurophysiological and neuroimaging literature and that our findings are more amenable to translation into clinical models. The advantage of the logistic regression used in our study is that it is simple, needs fewer computational resources (time and memory) than the more sophisticated machine learning approaches, can be run in most programming languages, and is easier to understand with regard to the relationship between the predictors and the dependent variables, while also yielding high performance.

We observed here a number of specific relationships between brain oscillations and symptom profiles. Though these findings were made in a data-driven, unbiased manner, they correspond with the prior literature. First, we found that the connectivity of the bilateral cuneus region in the *α* brain rhythm is a significant hub predicting anxiety symptoms. The physiological effects are strongest using the interference processing task (Middle Fish). These findings map onto a broader array of findings linking changes in posterior *α* oscillations (cuneus and precuneus) with various anxiety disorders. Abnormalities in visual-evoked *α* power have been observed in PTSD [67], social anxiety [9], and anxiety [68,69]. Posterior *α* activity may be related both to DMN activity [70], as well as to functional network changes between the DMN and the salience network, which is often a marker of hyper-vigilant behaviors.

We observed that the temporal pole and frontal pole in the *θ* rhythm were strong classifiers for depression symptoms, with the interference processing task and emotional interference task contributing the most to this classification. The temporal lobe and frontal polar regions correspond to brain areas wherein reduced effective connectivity is found within depressed individuals [71]. Prior work has demonstrated that theta activity is a marker for depression [72,73,74] and *θ* rhythms are strongly linked with hippocampal/temporal lobe processes [75]. Temporal lobe structures are a key part of the hippocampal default-mode-network, which is heavily implicated in depression [76,77]. Thus, it is possible that in our set of cognitive assessments, the temporal lobe structures were the strongest markers of altered DMN-like activity.

The right bank’s superior temporal sulcus in the *α* rhythm best-predicted inattention symptoms, with the interference processing and emotional processing tasks being particularly important for this prediction. The superior temporal sulcus has been previously implicated in audiovisual/multimodal processing [78] and social attention [79], and is a key part of the ventral attention network [80], which is involved in orienting to sudden/unexpected external information. The ventral attention network (and temporoparietal activity in particular) is abnormal in ADHD [81], with inattention symptoms specifically linked with abnormal patterns of ventral attention network connectivity [82,83,84].

Further, we observed that the left supramarginal area in the *β* rhythm significantly predicted hyperactivity, with the two go-nogo task blocks being the most significant contributors to this prediction. Lower activation of the supramarginal area during hyperactivity has also been evidenced in prior studies [85], and we find these activations primarily as event-related desynchronizations in the *β* band [29]. Overall, the predominance of the interference processing task in predicting symptoms is interesting given that such processing is fundamental to cognition and is also susceptible to significant plasticity throughout the lifespan [86,87].

With regards to the major limitations of this study, here we used a single dataset administering five different cognitive tasks to 97 healthy adult subjects. In this limited dataset, we did not find that the original feature set was able to accurately predict mild/moderate mental health symptoms. This could be due to an insufficient number of subjects and/or insufficient differentiation of symptoms across subjects. Hence, this work needs to be replicated and improved upon using data from clinical populations, and as of now is best interpreted as a methodological exercise. A future repeat-visit design of such a study can also shed light on the test–retest reliability, or the time-varying nature of the predictive methods. Performing group comparisons between tasks, frequencies, and ROIs is beyond the scope of the current study and is also planned for future work.

Overall, the present study shows the preliminary utility of source-reconstructed EEG spectral signals evoked within multiple cognitive task contexts in the prediction of psychiatric symptoms. It lays the foundation for replicating such work in clinical data followed by verification of the neural predictors using closed-loop approaches applied within cognitive contexts [4,88,89,90,91,92].

## Figures and Tables

**Figure 2 sensors-22-03116-f002:**
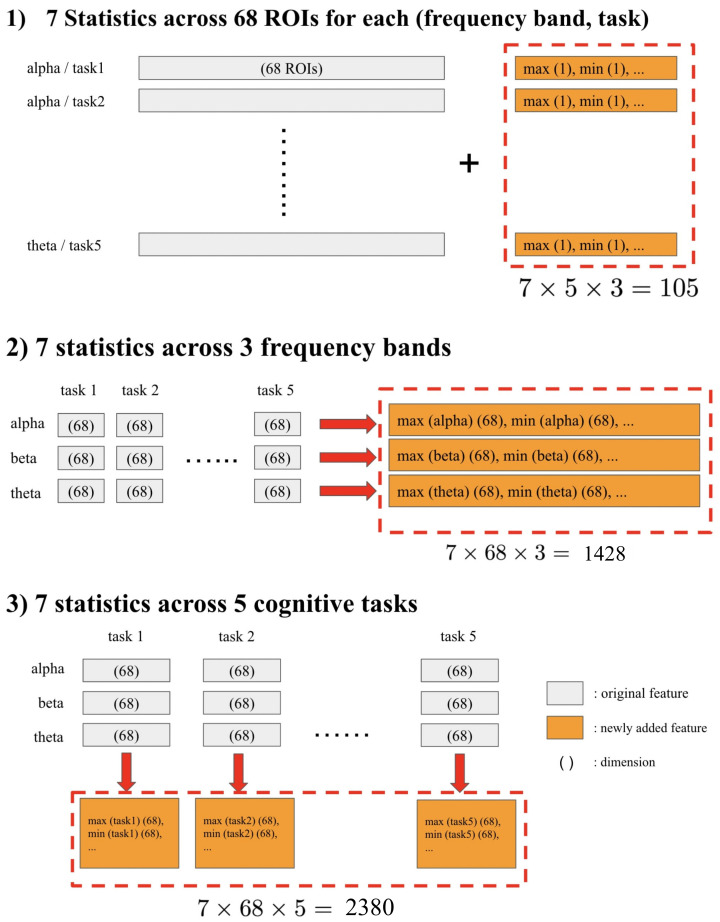
Various statistical measures added as new features.

**Figure 3 sensors-22-03116-f003:**
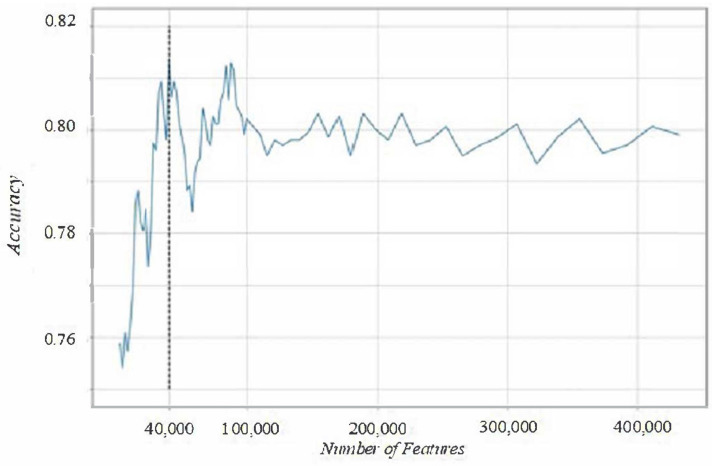
Prediction accuracy of logistic regression for symptom scores for top-k augmented features in chi-square statistic. As number of features (horizontal axis) increases, the accuracy initially increases rapidly and then reaches a plateau around 40,000 features (black vertical dash-line).

**Figure 4 sensors-22-03116-f004:**
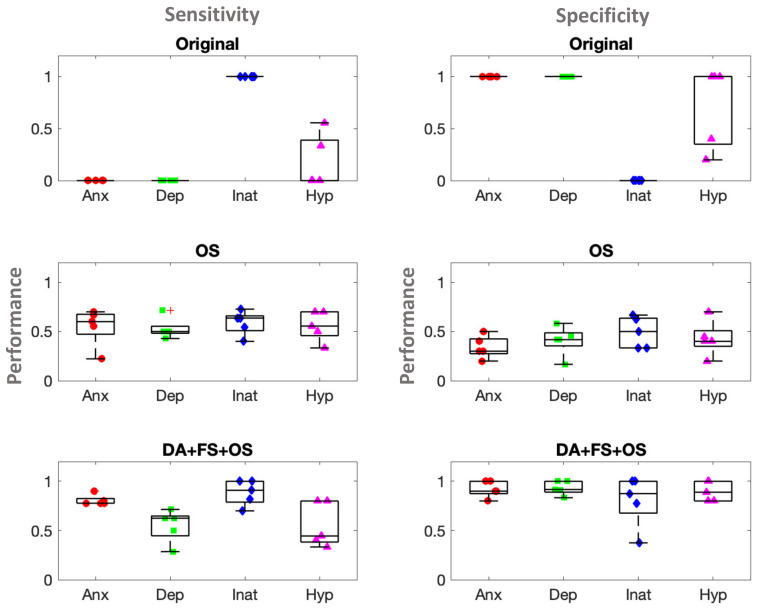
Comparison of performance of logistic regression with and without modification of the original dataset. Bar plots show comparison of prediction performance, sensitivity (**left column**), and specificity (**right column**) of 4 mental health symptoms (i) anxiety, (ii) depression, (iii) inattention, and (iv) hyperactivity, by logistic regression applied to different datasets.Original: original dataset (**first row**); OS: over-sampled (SMOTE and adding Gaussian noise) dataset (**second row**); DA + FS + OS: dataset that underwent all three processes (**third row**), data augmentation (increase of feature), feature selection (reduction of feature) based on chi-square statistic, and oversampling (SMOTE and adding Gaussian noise).

**Figure 5 sensors-22-03116-f005:**
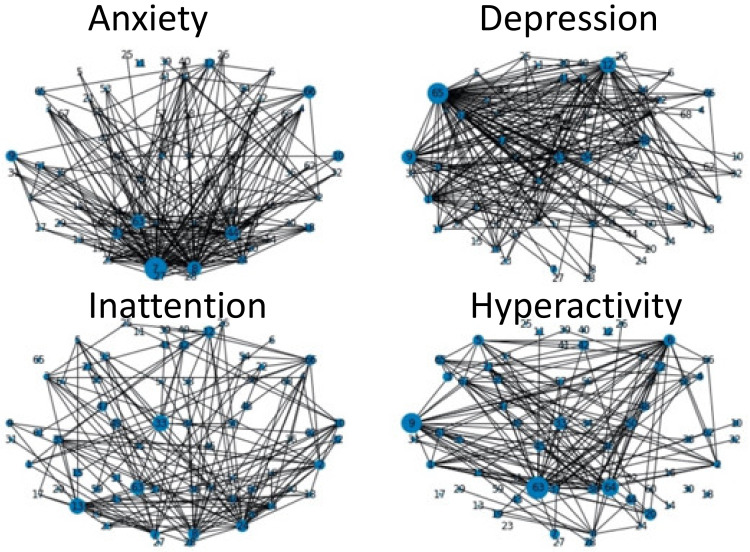
Topographical plot showing “strength” of ROI connections of relevance to mental health disorders, anxiety (**upper left**), depression (**upper right**), inattention (**bottom left**), and hyperactivity (**bottom right**). Size of each vertex corresponds to sum of chi-square statistic of the vertex. An edge is created when the sum of chi-square statistic of the two vertices that share the edge exceeds threshold: 100. The thickness of all edges is the same.

**Figure 6 sensors-22-03116-f006:**
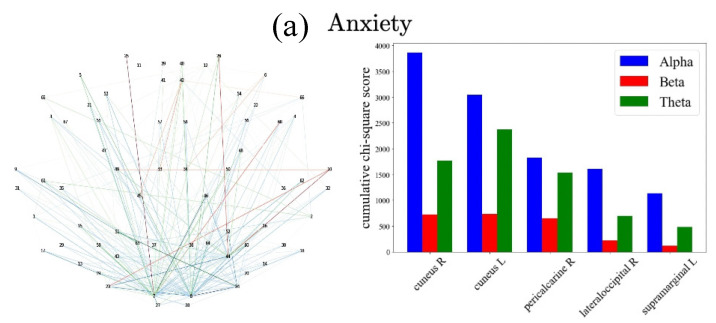
Frequency bands with highest cumulative chi-square scores across ROIs. (**left**) Topographical plot of ROIs whose edges are colored to represent the frequency band with the largest cumulative chi-square score. (**right**) The cumulative chi-square value of each frequency band in the top 5 ROIs.

**Figure 7 sensors-22-03116-f007:**
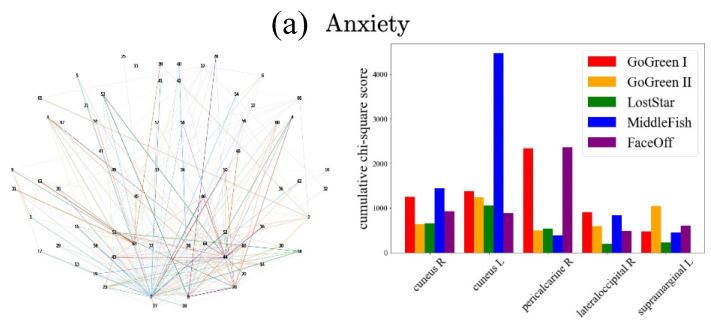
Tasks with highest cumulative chi-square scores across ROIs. (**left**) Topographical plot of ROIs whose edges are colored to represent the cognitive task with the largest cumulative chi-square score. (**right**) The cumulative chi-square value of each of the 5 tasks in the top 5 ROIs. Go Green I is a go-nogo selective-attention task, Go Green II is a go-nogo response inhibition task, Lost Star is a working memory task, Middle Fish is an interference processing task, and Face Off is an emotional interference processing task.

**Table 1 sensors-22-03116-t001:** Top 10 features with highest chi-square statistic for predicting anxiety. Augmented features consist of, at most, 2 original features and incur, at most, two transformations (product of pairs and log transform). This table shows details of top 10 features in terms of chi-square statistic.

			Original Feature 1			Original Feature 2		
Rank	Log	Product	Freq Band	Task	ROI ID	Freq Band	Task	ROI ID
1	Yes	Yes	Alpha	2	19	Alpha	2	41
2	No	Yes	Alpha	2	19	Alpha	2	41
3	Yes	Yes	Alpha	5	7	Alpha	5	43
4	No	Yes	Theta	2	39	Theta	5	7
5	No	Yes	Alpha	5	7	Alpha	5	43
6	Yes	Yes	Theta	1	45	Theta	5	6
7	Yes	Yes	Theta	2	39	Theta	5	7
8	Yes	Yes	Theta	1	45	Theta	5	7
9	No	Yes	Theta	5	9	Theta	5	19
10	No	Yes	Theta	1	7	Theta	5	22

**Table 2 sensors-22-03116-t002:** ROIs with top 5 centrality scores. Top 5 ROIs in terms of two types of centralities (closeness centrality and betweenness centrality) for each mood symptom. The values of centrality are computed based on weighted graphs in which vertices, edges, and weights correspond to ROIs, a pair of ROIs, and chi-square statistic of augmented features consisting of two ROIs of two original features.

ID	Name	Closeness	Betweenness
**Anxiety**			
8	cuneus R	6.272	0.339
7	cuneus L	6.171	0.268
44	pericalcarine R	5.866	0.145
24	lateraloccipital R	5.767	0.116
63	supramarginal L	5.356	0.135
**Depression**			
65	temporalpole L	2.281	0.220
9	entorhinal L	2.279	0.245
42	parstriangularis R	2.216	0.144
12	frontalpole R	2.212	0.145
33	paracentral L	2.157	—
67	transversetemporal L	—	0.099
**Inattention**			
2	bankssts R	3.357	0.271
24	lateraloccipital R	3.315	0.135
13	fusiform L	3.293	0.147
63	supramarginal L	3.187	0.119
35	parahippocampal L	3.118	0.098
**Hyperactivity**			
63	supramarginal L	3.240	0.194
8	cuneus R	3.202	0.183
9	entorhinal L	3.135	0.140
44	pericalcarine R	3.005	0.142
67	transversetemporal L	2.839	0.080

## Data Availability

The data is available on reasonable request to the corresponding author.

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
