# Peer review of "Utility of Cognitive Neural Features for Predicting Mental Health Behaviors"

_sensors, 2022, doi:10.3390/s22093116_

Round 1

Reviewer 1 Report

This study aimed to utilize classify healthy and mild to moderate disorder using cognitive neural markers using electroencephalography (EEG).  I have the following suggestions.

  • Which novelty do authors claim for this study, although similar concepts has been proposed earlier?
  • Introduction should be improved and should cover the state-of-art neurological biomarkers for mental workload and disorder prediction, disease classification. Machine-learning-based biosignal-based classification studies should be explored in the Introduction section. Authors should explore state-of-art biosignal application in mental workload, disease prediction, mentioning the references, doi:10.3390/s21216985.
  • The description of scientific information is very poor and difficult to follow. Scientific explanations are not clear, and logics are not well-connected to each other. Authors should review whole manuscript with an expert in neuroscience domain.
  • Few abbreviations are missing in this manuscript. Authors should recheck the manuscript.
  • In line 72 and 92, “Dataset Acquisition adapted from [9]”, “Experimental Design, adapted from [9]” are inappropriate expressions. Please exclude “adapted from [9]” from sub-title and explain in the description.
  • EEG is highly sensitive to the powerline, muscular and cardiac artifacts. In EEG data preprocessing, authors need to mention how you handle AC power, ECG, movement, and EMG artifacts in EEG signals. Line 125-126, authors mention to filter >45Hz signal for muscular artifact removal. How authors can be confirmed that muscular artifact exists only >45 Hz? Authors tried to use independent component analysis (ICA) to discriminate ECG, EMG, motion artifact from EEG signals. Authors didn’t mention about recording EEG, ECG, EMG signals of participants, although ICA requires recording of simultaneous EEG, ECG, EMG recording to isolate signals.
  • Authors should add a Figure of signal processing protocol with artifact removal used in this study.
  • Authors should report the statistical results of EEG event-related spectral perturbations in the theta, alpha and beta frequency bands in five tasks.
  • EEG Frequency-domain features are responsive for mental workload, type of motor workload and brain stimulations, such as, doi:10.3390/brainsci11070900. Authors must listed-up and make discussion on the advantages and drawbacks of their proposed method with other studies adding a discussion section.
  • From the writing point of view, the manuscript needs to be checked for typos and the English language should be improved.

Reviewer 2 Report

this work present the preliminary utility of source-reconstructed EEG spectral signals evoked within multiple cognitive task contexts in prediction of psychiatric symptoms. However, this is based in a sample of healthy individuals.

I found merit in the work and methods used but many other details and tests need to be included to improve the manuscript. 

I cannot recommend this to publication at this stage but let here a few comments I hope the authors can use to improve the paper:

Page1Line23: healthy people have symptoms but have not the disease?

P2L54: more details about other studies trying the same approach or testing similar hypotheses are needed. The hypothesis and novelty of the work are not clear.

P2L69: this is very confusing to me. Healthy people scored above the threshold for disease-related scales? So they were not healthy?

P2L88: these five would be the control group to compare the results of the healthy group. why not test them? I would test the machine learning on them and other diagnosed subjects as control.

P3L105: a detailed description of these tasks is missing.

P4L131: 24channels are very limited to perform source localization. This needs to be better justified and authors should show this number of channels is enough. No source map is shown in the results.

P5L184: which software was used? Further details are needed in the methods to make it possible for replication.

P6L225: I might be wrong, but the authors should look for reducing the number of features. To do that, we first increase them? Please explain.

P8L281: 5iterations: isn't this number too low?

P12L414: please  include a section with the ERPs or TF results of each task in the time-window used for the classification as well as the source localization maps.

fig4: please report boxplot with scatter points.

L472: include these labels over each connectivity map. it makes the figure easier to understand.

L503: please report statistical significance results of the comparison between tasks/frequency/rois.

overall I did not find the results of specificity and sensitivity significance tests. What is the chance level? 

Reviewer 3 Report

The manuscript by Kato et al present a very interesting approach and important results. Overall, manuscript is well thought and nicely written. My only concern is that Introduction does not introduce to the state of the art in the field - please overview what is know so far in relation to the object and clearly state the novelty of the current work.

Round 2

Reviewer 1 Report

Thanks for addressing the comments.

Author Response

We thank the reviewer for finding merit and interest in our work, and for their invaluable guidance to improve our manuscript.